# SelfVC: Voice Conversion With Iterative Refinement using Self Transformations

## Abstract

We propose SelfVC, a training strategy to iteratively improve a voice conversion model with self-synthesized examples. Previous efforts on voice conversion focus on factorizing speech into explicitly disentangled representations that separately encode speaker characteristics and linguistic content. However, disentangling speech representations to capture such attributes using task-specific loss terms can lead to information loss by discarding finer nuances such as accent and emotion of the original signal. In this work, instead of explicitly disentangling attributes with loss terms, we present a framework to train a controllable voice conversion model on entangled speech representations derived from self-supervised learning (SSL) and speaker verification models. First, we develop techniques to derive prosodic information from the audio signal and SSL representations to train predictive sub-modules in the synthesis model. Next, we propose a training strategy to iteratively improve the synthesis model for voice conversion, by creating a challenging training objective using self-synthesized examples. In this training approach, the current state of the synthesis model is used to generate voice-converted variations of an utterance, which serve as inputs for the reconstruction task, ensuring a continuous and purposeful refinement of the model. We demonstrate that incorporating such self-synthesized examples during training improves the speaker similarity of generated speech as compared to a baseline voice conversion model trained solely on heuristically perturbed inputs. Our framework is trained without any text and is applicable to a range of tasks such as zero-shot voice conversion, voice conversion across different languages, and controllable speech synthesis with pitch and pace modifications. We conduct extensive comparisons against prior work and find that SelfVC achieves state-of-the-art results in zero-shot voice conversion on metrics evaluating naturalness, speaker similarity, and intelligibility of synthesized audio. [1]

## 1 Introduction

Deriving meaningful representations from speech has been a topic of significant interest because such representations can be useful for both downstream recognition and upstream speech generation tasks. While some techniques (Défossez et al., 2022; Eloff et al., 2019; Liao et al., 2022; Kumar et al., 2023) aim to compress speech into a data-efficient codec, another line of research has focused on disentangling the learned features into components such as speaker characteristics (voice or timbre), linguistic content (phonetic information) and prosodic information (pitch modulation and speaking rate) (Chou et al., 2019; Qian et al., 2019; Wu & Lee, 2020; Chen et al., 2021; Qian et al., 2022; Hussain et al., 2023). Representation disentanglement allows controllable speech synthesis by training a model to reconstruct the audio from the disentangled features. During inference, the relevant disentangled representations can be modified for performing tasks like voice conversion (changing the speaker of an utterance) or changing the prosody.

To derive disentangled speech representations in a text-free manner, recent methods (Lakhotia et al., 2021; Polyak et al., 2021; Lin et al., 2021; Huang et al., 2022; Choi et al., 2021) have proposed to obtain speaker information from a speaker verification model and linguistic content information from the output of models trained using self-supervised learning (SSL) (Baevski et al., 2020; Gulati et al., 2020). While the representations obtained from SSL models are highly correlated with phonetic

---

[1] Audio examples and demo: `https://selfspeechsynthesis.github.io/`

information, they also contain speaker information (Huang et al., 2022; Hussain et al., 2022). To remove speaker information from the SSL model outputs, some techniques utilize an information bottleneck approach such as quantization (Polyak et al., 2021; Lakhotia et al., 2021; Gu et al., 2021). Alternatively, several researchers have proposed training strategies that employ an information perturbation technique to eliminate speaker information without quantization (Qian et al., 2022; Choi et al., 2021; 2023; Hussain et al., 2023). Notably, for training synthesizers, NANSY (Choi et al., 2021) and NANSY++ (Choi et al., 2023) propose to heuristically perturb the voice of a given utterance with hand-engineered data augmentations, before obtaining the output from the SSL model. To reconstruct the original audio accurately, the synthesizer is forced to derive the speaker characteristics from the speaker embedding since the speaker information in the SSL model's output is perturbed. While such techniques are effective, heuristic voice perturbation algorithms based on pitch randomization and formant shifting represent a very limited set of transformations. We hypothesize that such training strategies can be improved by utilizing neural network-generated augmentations.

In this work, we propose SelfVC, a learning framework to automatically generate diverse data transformations during training and enable controllable speech synthesis from imperfectly disentangled but uncompressed speech representations. First, we propose a feature extraction pipeline to derive SSL representations, speaker embeddings and prosodic information from a given audio signal. Next, we design a synthesis model to reconstruct a given utterance from the SSL features and speaker embedding, while using the fundamental frequency contour and duration as targets for training intermediate submodules. Finally, to train an effective voice conversion model, we propose a training strategy that utilizes the synthesis model itself to create challenging voice-converted transformations of a given speech utterance. At any given training iteration, the current state of the synthesis model is used to transform the input SSL features and the model is updated to minimize the reconstruction error of the original utterance.

All the components in our framework are trained in a text-free manner requiring only audio data. Once trained, our framework can be used for tasks such as zero-shot voice conversion, audio reconstruction with pitch and duration modulation as well as multilingual voice conversion across languages outside of the training set. On metrics evaluating speaker similarity, intelligibility and naturalness of synthesized speech we demonstrate that our model outperforms previously proposed zero-shot voice conversion methods. The main contributions of our work are:

1. We develop a training strategy using self transformations to train a voice conversion model on imperfectly disentangled representations, resulting in considerable improvement in speaker similarity metrics as compared to a model trained only with heuristic transformations.

2. We propose techniques to derive prosodic information from uncompressed SSL feature vectors and use the derived information to train a controllable synthesizer that can either mimic the prosody of a source utterance or adapt the prosody given a target speaker.

3. Our models are trained in a text-free manner and independent of phonetic posteriograms, hence making it simple and efficient to scale up the training data, including other languages.

4. SelfVC achieves state-of-the-art results in zero-shot any-to-any voice conversion in English. When fine-tuned on a few hours of multi-lingual data, SelfVC outperforms prior voice conversion methods on the cross-lingual voice conversion task.

## 2 RELATED WORK

**Voice conversion:** Voice conversion is the task of modifying an utterance of a source speaker to match the vocal qualities of a target speaker. Traditionally, voice conversion models were trained as a speech-to-speech translation system on a parallel dataset containing multiple speakers saying the same utterance (Sun et al., 2015; Chen et al., 2014). More recently, voice conversion systems have been developed by training neural synthesizers to reconstruct speech from disentangled representations describing linguistic content and speaker characteristics (Qian et al., 2019; Chou et al., 2019). For example, (Sun et al., 2016; Tian et al., 2018) have utilized pre-trained automatic speech recognition (ASR) and speaker verification (SV) models to disentangle content and speaker information respectively. The predicted text or phonetic posteriogram (PPG) obtained from the ASR model is taken as the content representation. However, such voice conversion systems have limitations: 1) Training such systems requires transcribed speech data and the synthesis is limited to the language the model

is trained on. 2) Text and PPG do not capture all linguistic features such as accent, expressions, emotions or speaker-independent style resulting in neutral-sounding synthesized speech.

To derive linguistic content in a text-free manner, some prior works have utilized SSL based models. However, as noted by prior work (Polyak et al., 2021; Huang et al., 2022), SSL model outputs do not necessarily separate speaker and content information. One line of research (Polyak et al., 2021; Lee et al., 2021; Lakhotia et al., 2021; Gu et al., 2021) aiming to disentangle the speaker and content representations, proposes an information bottleneck approach to quantize SSL model outputs thereby limiting the information to only capture the content or pseudo-text of the audio. However, the loss of information during such a quantization approach leads to sub-optimal reconstruction quality. Moreover, information bottleneck by itself does not guarantee disentanglement.

Addressing the limitations of information bottleneck approaches, researchers have proposed training strategies based on heuristic transformations. For example, in ContentVec (Qian et al., 2022) and ACE-VC (Hussain et al., 2023), while training the SSL-based feature extractor model, the audio is transformed using pitch-shift transformation and the SSL model is trained to output similar representations for the original and transformed audio. Alternatively, in NANSY (Choi et al., 2021), the transformations are applied while training the synthesizer, i.e. the synthesizer is tasked to reconstruct the original audio from the speaker embedding of the original audio and the SSL features of audio perturbed using transforms such as formant-shift, pitch-randomization and randomized frequency shaping. Although these heuristic transformations serve as a reasonable proxy for voice conversion methods, we hypothesize such methods can be greatly improved by utilizing the voice conversion system itself to generate more diverse input transformations.

**Transformation invariant representation learning:** In unsupervised representation learning, prior work has investigated methods to learn representations that are invariant to various input transformations (Bachman et al., 2019; Misra & Maaten, 2020). Several techniques addressing this challenge utilize domain-specific and hand-engineered data augmentation methods for training transformation invariant representation encoders (Chen et al., 2020; Caron et al., 2020; Tian et al., 2020; Grill et al., 2020; Misra & Maaten, 2020). Stochastic data augmentation in the image domain such as cropping, rescaling, shifts in brightness and recoloring have been popularly used to learn robust representations for image classification tasks (Chen et al., 2020; Tian et al., 2020). More recently, (Tamkin et al., 2021) proposed to train generative models to produce diverse views from a given input by adding a bounded perturbation. Their results demonstrate that neural generative models can produce a more diverse set of input distortions (compared to hand-engineered augmentations) without requiring domain-specific knowledge. While these techniques have proven valuable for learning transformation-invariant representations in downstream recognition tasks, their applicability in upstream generative tasks remains unexplored. In our work, we develop a novel framework for training a controllable synthesis model using self-generated input transformations. In contrast to previous ideas, we do not introduce additional networks for data augmentation but utilize the synthesizer model itself to generate diverse input transformations.

## 3 APPROACH

Our goal is to design a voice conversion framework that can modify the voice of a given utterance, while also providing control over the prosody of the synthesized speech. To this end, our framework consists of two main components: (1) A feature extractor that derives content (linguistic features), speaker embedding and prosody information from a given speech utterance (Section 3.1); and (2) A synthesizer model that reconstructs the audio from the derived representations (Section 3.2). To allow controllable synthesis from imperfectly disentangled representations, we propose a training strategy that challenges the model to reconstruct the audio from self-generated perturbations of the content representation (Section 3.3). Specifically, we train the model to reconstruct the audio from the content representation of a heuristically modified or self transformed audio, while preserving the speaker and style representations. The content and speaker encoder networks remain frozen during synthesis model training. Figure 1 provides an overview of our voice conversion framework and the synthesizer training procedure.

### 3.1 FEATURE EXTRACTION

The overview of the feature extraction pipeline is shown in Figure 2 (a). We derive the following features from an audio signal to train our synthesis models.

**Content Embedding:** We define content as a temporal feature that encodes the linguistic information of a given speech utterance. We use the output of the Conformer-SSL (Gulati et al., 2020) model ($G_c$) as the content representation of speech ($z$). The Conformer-SSL model is a convolution-augmented transformer architecture that is trained to reconstruct the masked areas of the mel-spectrogram on English speech data, using contrastive and masked language modelling (MLM) losses (Refer to Appendix A.2 for model details). Given a speech utterance as a sequence of mel-spectrogram frames $x = x_1 \ldots x_T$, the Conformer-SSL model outputs a temporally downsampled sequence of feature vectors $z = G_c(x) = z_1 \ldots z_{T'}$. While $z$ primarily encodes phonetic information, it also encompasses speaker and prosodic information. We explain our approach to address this challenge for training a voice conversion model in Section 3.3.

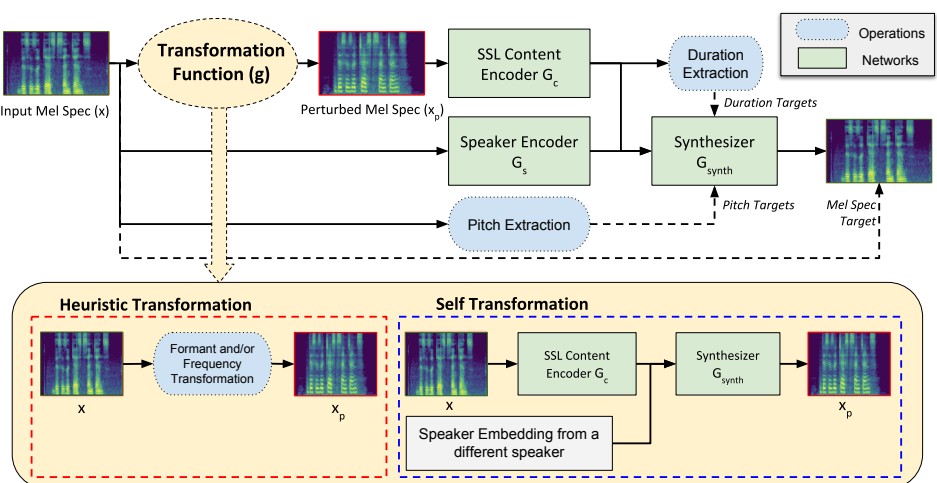

Figure 1: SelfVC Overview: The synthesizer $G_{synth}$ is trained to reconstruct the mel-spectrogram from SSL-based content representation of a transformed audio and speaker embedding of the original audio. The transformation function is either a heuristic transform or a voice-converted audio generated using self-synthesis with a different speaker embedding.

**Duration:** Duration or rhythm characterizes the speaking rate at a granular level, that is, how long the speaker vocalizes each phoneme of a given utterance. Accurate modelling of rhythm during synthesis is important to capture the nuances between the different speakers, accents and emotions. Since SSL representations have a high correlation with phonemes (Baevski et al., 2020; Gulati et al., 2020), we conjecture that if a phoneme is emphasized in an utterance, the consecutive content vectors at the corresponding timesteps will have high similarity. Therefore, we group together consecutive content vectors with cosine similarity higher than a threshold $\tau$, and set the target duration for the averaged vector as the number of grouped vectors multiplied by the duration of a single vector. That is, we process the content representation $z = z_1 \ldots z_{T'}$ into a duration-augmented content representation $z' = z'_1 \ldots z'_{\hat{T}}$ and $d' = d'_1 \ldots d'_{\hat{T}}$ where $\hat{T} \leq T'$ and $d'_t$ represents the duration of $z'_t$. This similarity based grouping approach is analogous to prior approaches (Lee et al., 2021; Qian et al., 2021). We refer readers to Algorithm 1 in the Appendix which details our approach to obtain $z', d'$ and highlights key differences with prior methods.

**Speaker Embedding:** The speaker embeddings in our setup are derived from the TitaNet (Koluguri et al., 2022) speaker verification model ($G_s$). The speaker verification model is trained to distinguish different speakers and generates similar embeddings for utterances from the same speaker. The output from the TitaNet speaker verification model is a 192 dimensional speaker embedding $s = G_s(x)$. We provide more details on this model in the Appendix A.2.

**Pitch Contour:** The pitch contour $p$ is derived from the fundamental frequency $f_0$ contour of the speech signal that represents the prosodic modulations over time. The raw values in the fundamental frequency contour (derived from PYin algorithm (Mauch & Dixon, 2014)) are speaker-dependent, therefore $f_0$ is not strictly disentangled from the speaker information. To ensure that the pitch contour only encodes the intonation and not the speaker identity, we normalize $f_0$ using the mean ($f_{mean}$) and standard deviation ($f_{std}$) of all pitch contours of the given speaker. That is, $p = (f_0 - f_{mean})/f_{std}$.

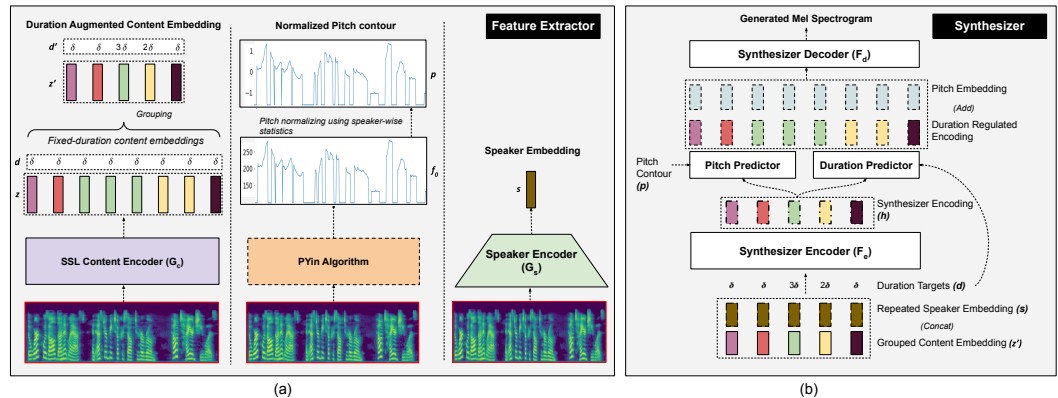

Figure 2: (a) Feature Extraction: The feature extractor derives the duration augmented content information from an SSL model, pitch contour using PYin algorithm and speaker embedding from a speaker verification model. (b) Mel Spectrogram Synthesizer: reconstructs the mel-spectrogram from the derived features.

## 3.2 SYNTHESIZER

The task of the synthesizer is to first reconstruct the ground-truth mel-spectrogram from the extracted speech representations and then vocode the mel-spectrogram into a listenable audio waveform. For vocoding, we use a HiFiGAN (Kong et al., 2020) vocoder, which is trained separately on spectrogram and waveform pairs from a multi-speaker dataset.

Our mel-spectrogram synthesizer $G_{synth}$ is composed of two feed-forward transformers $F_e$ and $F_d$ and intermediate modules to predict the duration and pitch contour similar to (Łańcucki, 2021) but operates on the grouped content representation $z' = z'_1 \ldots z'_{T'}$ instead of text. The speaker embedding $s$ is repeated across all time-steps and concatenated with each $z'_t$ to be fed as input to the first feed-forward transformer $F_e$. The hidden representation from $F_e$ is then used to predict the duration and pitch contour, that is: $h = F_e(z', s)$; $\hat{y_d} = DurationPredictor(h)$, $\hat{y_p} = PitchPredictor(h)$. The pitch contour is projected and averaged over each time-step of the hidden representation $h$ and added to $h$ to get $k = h + PitchEmbedding(p)$. Finally, $k$ is discretely upsampled as per the ground-truth duration $d'$ and fed as input to the second transformer $F_d$ to get the predicted mel-spectrogram $\hat{y} = F_d(DurationRegulation(k, d'))$. Our model is trained to optimize three losses — mel-reconstruction error, pitch prediction error and duration prediction error such that

$$L_{synth} = \|\hat{y} - y\|_2^2 + \lambda_1 \|\hat{y_p} - p\|_2^2 + \lambda_2 \|\hat{y_d} - d'\|_2^2 \tag{1}$$

During inference, we can use either the predicted pitch and duration, in which case the prosody is derived from both the content and speaker embeddings; or we can mimic the prosody and speaking rate of the source utterance by using ground-truth duration and pitch information.

## 3.3 SYNTHESIZER TRAINING: ITERATIVE REFINEMENT USING SELF TRANSFORMATIONS

While the mel-spectrogram can be accurately reconstructed from a synthesizer trained using the objective given by Equation 1, during inference, we cannot effectively modify the voice of a given utterance. This is because the content representation $z'$ is not strictly disentangled from the speaker information. To address this challenge, past works (Choi et al., 2021; 2023), have proposed an information perturbation based training strategy as follows: Instead of feeding the content embedding of the original audio as the input, the audio is perturbed to synthetically modify the speaker characteristics using formant-shifting, pitch-randomization and randomized frequency shaping transforms to obtain $x_p = g_{heuristic}(x)$. Next, the content embedding is derived from the perturbed audio $z' = G_c(x_p)$, while the speaker embedding is still derived from the original audio $s = G_s(x)$. The network is then tasked to reconstruct the original audio from $z'$ and $s$. While heuristically perturbed content representations play a crucial role in enhancing the synthesizer model's attention towards the speaker embedding, they are limited in terms of the range of transformations they can introduce. Heuristic transformations represent only a subset of the potential natural variations that can occur during voice conversion.

To expand on the heuristic set of transforms, we propose to utilize the synthesizer model itself to generate a voice-converted variation of a given utterance $x$. That is, given a synthesizer model $G_{synth}^i$ trained until training iteration $i$, we obtain a self transformed audio for iteration $i + 1$ as:

$$x_p = g_{self}(x) = G_{synth}^i(G_c(x), s') \tag{2}$$

where $G_c(x)$ is the content embedding of the original audio $x$ and $s'$ is the speaker embedding obtained from an utterance $x'$ of a different randomly selected speaker, that is, $s' = G_s(x')$. The content embedding input for the training step $i + 1$ is then derived as $z' = G_c(x_p)$.

Self transformations not only provide a more diverse set of transformations but also present an increasingly challenging reconstruction task for the synthesizer, as its voice conversion capabilities improve with each training iteration. Figure 1 demonstrates the proposed self transformation training strategy. In our experiments, we begin self transformations after $100k$ mini-batch iterations of training with heuristically modified audio. Thereafter, we get a reasonable initialization for a voice conversion model, and we start using self transformations to obtain $x_p$ as per Equation 2.

## 4 EXPERIMENTS

### 4.1 DATASET AND TRAINING

The Conformer-SSL model used as the content encoder is pretrained on $56k$ hours of unlabelled English speech from the LibriLight (Kahn et al., 2020) corpus sampled at 16 KHz. We fine-tune the Conformer-SSL model (using self-supervision with contrastive and MLM loss) on the *train-clean-360* subset of LibriTTS (Zen et al., 2019) dataset with audio sampled at $22050 Hz$ to make the model compatible with the mel-spectrogram representation of the synthesizer. For both the content encoder and synthesizer, we use $80$ bands for mel spectrogram with the FFT, window, and hop size set to $1024$, $1024$, and $256$ respectively. We fine-tune the Conformer-SSL on this revised spectrogram representation for 50 epochs with a batch size of 32 using the AdamW optimizer with a fixed learning rate of $5e - 5$ and $\beta_1 = 0.9, \beta_2 = 0.99$. Fine-tuning takes around 50 hours on a single NVIDIA RTX A6000 GPU.

For our primary experiments, the mel-spectrogram synthesizer and the HifiGAN vocoder are also trained on the train-clean-360 subset of the LibriTTS dataset which contains 360 hours of speech from $904$ speakers. We train three variants of the mel-spectrogram synthesizer:
**1. Baseline–NoTransform** is trained to simply reconstruct the mel-spectrogram from the embeddings of the given utterance without any information perturbation procedure.
**2. Baseline–Heuristic** is trained to reconstruct the mel-spectrogram from the content embedding of the heuristically perturbed utterance and the speaker embedding of the original utterance. We employ two transforms $g_1, g_2$ proposed in (Choi et al., 2021). $g_1$ perturbs formant, pitch, and frequency response and $g_2$ perturbs formant and frequency response while preserving pitch. The hyperparameter details of these transformations are provided in the Appendix A.3.
**3. SelfVC** is first trained in the same way as Baseline–Heuristic for the first $100k$ mini batch iterations. Thereafter, we use the $g_{self}$ transformation procedure given by Equation 2.

All three variants of the synthesizer are optimized using an AdamW optimizer (Loshchilov & Hutter, 2019) with a fixed learning rate of $1e - 4$ and $\beta_1 = 0.8, \beta_2 = 0.99$ for 500 epochs with a batch size of 32. The threshold $\tau$ for duration extraction is set as $0.925$. The loss coefficients for the duration and pitch loss are set as $\lambda_1 = \lambda_2 = 0.1$. The training time for Synth (SelfTransform) model is around 5 days on 4 NVIDIA RTX A6000 GPUs. The HifiGAN vocoder is also trained on the train-clean-360 subset of the LibriTTS and the same vocoder is used across all three synthesizers. We point readers to Appendix A.2 for detailed architectures and implementation of various components.

### 4.2 EVALUATION METRICS

We encourage readers to listen to our audio examples linked in the footnote on the first page. Quantitatively, we evaluate the synthesized audio on the following aspects:

**Intelligibility:** For intelligibility, we transcribe the synthesized and original through and ASR and compute two error metrics between the transcriptions — Character Error Rate **(CER)** and Phoneme Error Rate **(PER)**. For CER, we transcribe the audio using the Quartznet (Kriman et al., 2020) ASR model. For multilingual evaluation, we compute the PER on the transcriptions obtained from the

pre-trained wav2vec2-Large-XLSR-53 ASR model which has been trained to recognize phonetic labels in multiple languages. (Xu et al., 2021). We also report the CER and PER between the predicted and ground truth transcripts of real data for reference in our Results.

**Speaker Similarity Metrics:** To evaluate speaker similarity to our target speaker, we compute the speaker embeddings of synthesized and real utterances using a separate pre-trained speaker verification model (Koluguri et al., 2020). Then we pair the synthesized and real utterances to create an equal number of positive (same-speaker) and negative (alternate-speaker) pairs for each target speaker to compute the Equal Error Rate **(SV-EER)**. We also report the mean cosine similarity between the positive pairs **(SV-SIM)**. Finally, we also ask human listeners to rate the speaker similarity of the generated and real utterance from the target speaker on a 5-point scale to obtain **Sim-MOS**.

**Naturalness (MOS):** We ask human listeners to rate the naturalness of each utterance on a 1 to 5 scale with 1 point increments. We include details of *MOS* and *SIM-MOS* evaluations in Appendix A.6

**Prosodic Similarity (GPE):** To evaluate prosodic similarity for the reconstruction task (Section 4.3), we compute error between the fundamental frequency contours of the original and synthesized audio. Specifically, we use Gross Pitch Error (GPE) (Chu & Alwan, 2009) to evaluate prosodic similarity.

Table 1: Reconstruction evaluation: Resynthesized speech from different synthesizers is evaluated for intelligibility (PER), speaker similarity (SV-EER) and prosodic similarity (GPE). Lower values are desirable.

| Dataset | Technique | Guided | | | Predictive | | |
|---|---|---|---|---|---|---|---|
| | | SV-EER ↓ | PER ↓ | GPE ↓ | SV-EER ↓ | PER ↓ | GPE ↓ |
| VCTK *(English)* *Seen Language* | Real Data | 3.1% | 9.8% | - | 3.1% | 9.8% | - |
| | Baseline–NoTransform | 4.6% | 5.1% | 7.8% | 4.7% | 5.4% | 11.1% |
| | Baseline–Heuristic | 4.3% | 4.9% | 7.9% | 4.5% | 5.3% | 11.1% |
| | SelfVC | 4.2% | 4.6% | 7.8% | 4.1% | 4.7% | 12.0% |
| CSS10 *(Multilingual)* *Unseen Language* | Real Data | 2.3% | 22.1% | - | 2.3% | 22.1% | - |
| | Baseline–NoTransform | 5.5% | 19.3% | 11.7% | 4.9% | 21.2% | 15.9% |
| | Baseline–Heuristic | 5.3% | 19.2% | 11.6% | 5.5% | 21.5% | 16.1% |
| | SelfVC | 4.1% | 19.5% | 10.8% | 4.8% | 21.1% | 16.8% |

## 4.3 RECONSTRUCTION

First, we evaluate how effectively our setup can reconstruct audio from the extracted representations for unseen utterances and speakers. Our synthesizers can operate in two modes during inference — *1) Guided:* In this scenario, we use ground truth pitch and duration information derived from the source utterance. *2) Predictive:* In this case, we use the predicted pitch and duration for synthesis. We conduct the reconstruction test on two unseen datasets — 1) We choose 200 utterances from the VCTK (Yamagishi et al., 2019) dataset (English) with 20 random utterances from each of the 10 speakers (5 random male and 5 random female speakers); 2) To evaluate performance on unseen languages, we choose 200 utterances from the CSS10 (Park & Mulc, 2019) dataset with 20 random utterances from each of the 10 unseen languages. The CSS10 dataset has a single speaker per language and contains at least 4 hours of speech per language. For both of these evaluations, we use the synthesizer models trained on the same dataset, i.e. train-clean-360 subset of LibriTTS (English). The synthesized speech is evaluated on the intelligibility, speaker similarity and prosodic similarity metrics.

As indicated by the results in Table 1, all three synthesizers achieve similar performance on the above metrics. This is expected since the speaker and content embedding are derived from the same utterance and all three synthesizers are trained for the reconstruction task. However, for controllable synthesis tasks such as voice conversion, we demonstrate that SelfVC considerably outperforms these baselines (Section 4.4). Since our model is trained in a text-free manner, we also see a promising generalization to unseen languages. The PER on CSS10 is higher than VCTK due to the larger phonetic vocabulary in non-English languages and the PER of the wav2vec2 model (Xu et al., 2021) being higher even on real data. For unseen languages, our synthesizers produce more intelligible speech in the guided mode, where the duration information of the source utterance is kept intact.

## 4.4 VOICE CONVERSION

To convert the voice of a given source utterance to a target speaker, we derive the content embedding from the source utterance and estimate the speaker embedding from the target speaker's audio and

feed both as input to the synthesizer. To compare our zero-shot voice conversion method against prior work, we choose utterances from the LibriTTS test-clean subset since it is an unseen dataset across all voice conversion methods. We randomly choose 10 target speakers (5 male and 5 female) and 20 source utterances from the remaining speakers to create 200 voice conversion trials for each technique and report the results in Table 2. For our primary evaluation, we use 10 seconds of speech from each target speaker to derive the speaker embedding. We split the 10 second target-speaker utterance into 2 second segments and estimate the speaker embedding as the mean speaker embedding across the segments. To be consistent with past work, we keep the duration of the source utterance unchanged during synthesis using duration guided mode and use predictive mode for pitch. We also evaluate the speaker-similarity performance for different amounts of target speaker data and present the results in Figure 3 of the Appendix. Additionally, we include voice conversion results on seen speakers and out-of-domain VCTK dataset in Appendix A.4.

Table 2: Comparison of different zero-shot voice-conversion techniques on speaker similarity, intelligibility and naturalness metrics (Section 4.2) on unseen speakers and source utterances from the test-clean LibriTTS subset using 10 seconds of target speaker audio.

| | *Speaker Similarity* | | | *Intelligibility* | | *Naturalness* |
|---|---|---|---|---|---|---|
| Technique | SV-EER ↓ | SV-Sim ↑ | Sim-MOS ↑ | PER ↓ | CER ↓ | MOS ↑ |
| Real Data | 2.6% | 0.61 | $4.36 \pm 0.08$ | 8.7% | 6.7% | $4.30 \pm 0.08$ |
| AdaIN-VC (Chou et al., 2019) | 28.7% | 0.36 | $2.62 \pm 0.07$ | 14.3% | 15.5% | $2.14 \pm 0.08$ |
| MediumVC (Gu et al., 2021) | 27.4% | 0.40 | $2.82 \pm 0.08$ | 27.7% | 29.1% | $2.51 \pm 0.07$ |
| FragmentVC (Lin et al., 2021) | 23.3% | 0.39 | $2.28 \pm 0.08$ | 27.0% | 31.1% | $2.42 \pm 0.07$ |
| S3PRL-VC (Huang et al., 2022) | 20.5% | 0.38 | $2.66 \pm 0.07$ | 12.5% | 9.6% | $2.81 \pm 0.08$ |
| YourTTS (Casanova et al., 2022) | 6.6% | 0.54 | $3.01 \pm 0.08$ | 8.4% | 4.9% | $3.49 \pm 0.07$ |
| ACE-VC (Hussain et al., 2023) | 6.6% | 0.49 | $3.29 \pm 0.09$ | 9.0% | 3.8% | $3.77 \pm 0.07$ |
| Baseline–NoTransform | 28.9% | 0.36 | $2.21 \pm 0.08$ | 5.5% | 1.9% | $3.95 \pm 0.05$ |
| Baseline–Heuristic | 6.0% | 0.53 | $3.65 \pm 0.07$ | 5.2% | **1.6%** | $3.97 \pm 0.06$ |
| **SelfVC** | **3.4%** | **0.58** | $\mathbf{3.74 \pm 0.07}$ | **5.1%** | **1.6%** | $\mathbf{4.06 \pm 0.06}$ |

**Effectiveness of Self Transformations:** We perform ablations to compare effectiveness of different input transformation techniques. As reported in Table 2, incorporating heuristic transformations during training (Baseline–Heuristic) improves speaker similarity of generated audio over a baseline that does not use any transformations (Baseline–NoTransform). The speaker similarity metrics (SV-EER, SV-Sim and Sim-MOS) further improve in SelfVC when we incorporate the self transformation based iterative refinement procedure (Section 3.3). Note that both the baseline techniques and the SelfVC approach use identical neural architectures and undergo training for the same number of epochs with consistent optimizer hyperparameters. It is interesting to note that while Baseline-NoTransform generates intelligible and natural-sounding audio, it clearly falls short on speaker similarity metrics indicating the importance of input transformation methods for voice conversion.

**Comparison against Prior Work:** Although we have conducted controlled experiments by varying input transformation techniques in our models, it is challenging to make similar comparisons with prior research due to disparities in vocoders, datasets, and compatibility of model architectures between synthesizers and vocoders. We use the official open-source implementations and model checkpoints of six previously proposed techniques. For a fair comparison, we evaluate all prior techniques on the same voice conversion trial pairs as our methods, using the same ASR and SV models for calculating CER, PER and SV metrics. While NANSY (Choi et al., 2023) is not officially open-sourced, our Baseline–Heuristic method closely follows the training strategy proposed in NANSY using the same hyperparameters for heuristic functions (Appendix A.3), incorporating more recent neural architectures for the synthesizer and feature extractors. As shown in Table 2, SelfVC outperforms previously proposed voice conversion models on all quantitative metrics. It is interesting to note that SelfVC trained on just the train-clean-360 subset of LibriTTS outperforms YourTTS which is trained on a much larger dataset comprising LibriTTS (train-clean-360, train-clean-100), VCTK and two additional languages (French and Portugese).

**Cross-lingual Voice Conversion:** For Cross-lingual voice conversion, we use the CSS10 dataset that contains speech utterances from 10 different languages. We consider three voice conversion scenarios: 1) **English to CSS10:** Source utterance is from the test-clean subset of LibriTTS (English) and target speaker is from the CSS10 dataset 2) **CSS10 to CSS10:** Source utterance from a language

Table 3: Results on cross-lingual voice conversion task in three scenarios considering different languages for source utterance and target speaker (described in Section 4.4). 10 seconds of target speaker utterance is used to derive speaker embedding. Lower SV-EER is desirable for higher speaker similarity and lower PER is desirable for more intelligible speech.

| Technique | English to CSS10 | | CSS10 to CSS10 | | CSS10 to English | |
|---|---|---|---|---|---|---|
| | SV-EER ↓ | PER ↓ | SV-EER ↓ | PER ↓ | SV-EER ↓ | PER ↓ |
| Real Data | 4.1% | 8.7% | 4.1% | 22.1% | 2.6% | 22.1% |
| AdaIN-VC (Chou et al., 2019) | 24.0% | 16.3% | 20.9% | 33.5% | 27.7% | 36.9% |
| MediumVC (Gu et al., 2021) | 29.3% | 28.4% | 28.1% | 43.3% | 29.8% | 45.7% |
| FragmentVC (Lin et al., 2021) | 25.6% | 29.9% | 25.0% | 42.4% | 26.4% | 44.1% |
| S3PRL-VC (Huang et al., 2022) | 31.5% | 12.9% | 30.1% | 32.9% | 23.1% | 35.1% |
| YourTTS (Casanova et al., 2022) | 13.2% | 8.5% | 12.5% | 23.6% | 9.7% | 28.0% |
| ACE-VC (Hussain et al., 2023) | 15.1% | 9.3% | 30.1% | 68.3% | 17.6% | 70.9% |
| Baseline–NoTransform (LibriTTS) | 24.6% | 5.3% | 30.9% | 23.7% | 26.2% | 23.8% |
| Baseline–Heuristic (LibriTTS) | 15.8% | 5.3% | 20.3% | 23.3% | 9.8% | 23.8% |
| **SelfVC (LibriTTS)** | 12.7% | **5.1**% | 18.4% | 23.7% | 7.0% | 25.4% |
| **SelfVC (LibriTTS + CSS10)** | **4.4**% | 6.0% | **4.4**% | **18.9**% | **4.8**% | **19.9**% |

in the CSS10 dataset and target speaker is from another language of CSS10. 3) **CSS10 to English:** Source utterance from a language in the CSS10 dataset and target speaker is from LibriTTS (English).

For *English to CSS10* we create 200 voice conversion trials considering 20 source utterances and 10 target speakers in CSS10. For *CSS10 to CSS10* and *CSS10 to English*, we generate 500 voice conversion trials each, considering 50 source utterances (5 each from the 10 languages) and 10 target speakers. We use 10 seconds of target speaker data across all experiments. We compare different voice conversion techniques on these trial pairs and present the results in Table 3.

In the *English to CSS10* experiments, SelfVC (LibriTTS), which is trained solely on train-clean-360 LibriTTS subset, outperforms baseline methods and prior work, achieving lower SV-EER and PER. It is interesting to note that SelfVC (LibriTTS) outperforms YourTTS, which is trained on a more extensive trilingual dataset as discussed above. For *CSS10 to CSS10* voice conversion, we observe a higher SV-EER and PER for SelfVC (LibriTTS) as compared to YourTTS. This is not very surprising, since YourTTS model was trained on multilingual speech data while SelfVC (LibriTTS) has only been trained on English speech. For *CSS10 to English* voice conversion, SelfVC (LibriTTS) outperforms all baselines and prior work. Interestingly, ACE-VC, which uses similar model architectures and the same training data as our setup, does not generate intelligible speech when the source utterance is from CSS10. This result indicates that the text-free nature of our model allows generalization to unseen languages.

To adapt SelfVC for new languages, we conduct fine-tuning of only the synthesis model on both LibriTTS (train-clean-360) and CSS10 utterances (using data other than the test trial pairs), which considerably improves SV-EER and PER for the SelfVC (LibriTTS + CSS10) model. The improvement in SV-EER is significant but not surprising since the 10 CSS10 speakers are now seen during training in the SelfVC (LibriTTS + CSS10) model. The improvement in PER is promising and demonstrates the effective adaptability of our model to different languages. We delve into details of the finetuning process and report the phoneme error rates for each of the 10 CSS10 languages in Appendix A.5.

## 5 CONCLUSION

We introduce a novel training strategy, SelfVC, that utilizes self transformations to train controllable synthesis models on imperfectly disentangled representations. Our results indicate a clear benefit of incorporating self-synthesized examples while training a voice conversion model, as shown by a significant improvement in speaker similarity metrics while keeping the model architecture unchanged. By deriving and modelling prosodic information during training, SelfVC allows for both fine-grained and high-level control over the prosody of the synthesized speech. SelfVC achieves SOTA results in zero-shot voice conversion for English and can be easily scaled to multiple languages in a text-free manner, outperforming prior approaches in cross-lingual voice conversion. We recommend future work to apply our training strategy in other data domains for creating controllable synthesis models.

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

# A  APPENDIX

## A.1  DERIVING DURATION-AUGMENTED CONTENT EMBEDDINGS

Given the output $z = G_c(x) = z_1 \dots z_T$ from the Conformer-SSL model, we group together consecutive feature vectors with high cosine similarity. That is, we maintain a running average of consecutive vectors with cosine similarity greater than a threshold $\tau$ and obtain the target duration for the averaged vector as the product of the number of grouped vectors and the duration of a single vector. The original duration $\delta$ of a single vector is $4$ mel-spectrogram frames or 46ms or raw audio. This procedure differs slightly from previous work (Lee et al., 2021) in that, instead of computing similarities between consecutive pairs of the original vectors, we now compare the average embedding of the current group with the next original embedding. Our temporal downsampling procedure is similar to (Qian et al., 2021) but we additionally maintain the durations of the grouped vectors to be used as targets for the duration predictor in our synthesizer. Our technique also differs from prior work (Kreuk et al., 2022; Maimon & Adi, 2022) that obtains duration/rhythm information from discrete SSL representations instead of the continuous vectors. Algorithm 1 details our grouping procedure to obtain duration-augmented content embeddings.

---

**Algorithm 1** Deriving duration-augmented content by grouping similar consecutive vectors

---

1: $z' \leftarrow [z_1]$ ▷ Initialize $z'$ with the vector from the first time-step in $z$
2: $d' \leftarrow [\delta]$ ▷ $d'_t$ represents duration of $z'_t$. $\delta$ represents duration of of each $z_t$ (i.e 4 frames)
3: $num\_grouped \leftarrow 1$ ▷ number of similar vectors grouped at the last processed time-step
4: **for** $t \leftarrow 2$ **to** $T'$ **do**
5:     **if** $CosineSimilarity(z_t, z'[-1]) > \tau$ **then** ▷ Group $z_t$ with the running group
6:         $z'[-1] \leftarrow (z_t + num\_grouped * z'[-1])/(num\_grouped + 1)$ ▷ Update average
7:         $d'[-1] \leftarrow \delta * (num\_grouped + 1)$
8:         $num\_grouped \leftarrow num\_grouped + 1$
9:     **else** ▷ Insert $z_t$ in a new group
10:         $z'.append(z_t)$
11:         $d'.append(\delta)$
12:         $num\_grouped \leftarrow 1$
13:     **end if**
14: **end for**
15: **return** $z', d'$

---

## A.2  MODEL ARCHITECTURE AND IMPLEMENTATION DETAILS

Our voice conversion comprises the following neural networks. Total number of parameters and inference latency for each model are listed in Table 4

**Conformer-SSL Model:** The Conformer-SSL model (Gulati et al., 2020) used in this work is a convolution-augmented transformer architecture that is trained to reconstruct the masked areas of the mel-spectrogram on English speech data, using contrastive and masked language modelling (MLM) losses. It is pre-trained on the LibriLight corpus which consists of 56k hrs of unlabeled English speech. The model consists of 18 layers, 8 attention heads and a hidden dimension of $512$. The output head of the Conformer model gives a $256$ dimensional encoding per timestep. The model temporally downsamples the input mel-spectrogram by a factor of $4$. With the STFT parameters used in our setup, each vector from the Conformer-SSL model corresponds to a contextualized representation of 46ms of audio.

**Speaker Verification TitaNet Model:** TitaNet (Koluguri et al., 2022) is based on a 1-D depthwise separable convolution architecture with Squeeze and Excitation layers that provide global context, followed by channel attention-based statistics pooling layer to map variable-length utterances to a fixed-length embedding. The TitaNet speaker verification model is trained using additive angular margin loss (Liu et al., 2017) on 3373 hours of speech from multiple datasets that span 16681 speakers. Comprising of 25.3 million parameters, the TitaNet model is designed to be parameter-efficient and achieves state-of-the-art results on the VoxCeleb-1 speaker verification benchmark with an EER of $0.68\%$. The output from this speaker verification model is a $192$ dimensional speaker embedding.

**Mel-spectrogram Synthesizer:** The spectrogram synthesizer takes as input the content and speaker embeddings and predicts the mel-spectrogram. The speaker and content embeddings derived from the Conformer-SSL and TitaNet models respectively are first projected to 256 dimensions each using a learnable linear layer. The projected speaker embedding is then repeated across all time-steps and concatenated with the projected content embeddings. The synthesizer is a FastPitch (Łańcucki, 2021) based model that contains two feed forward transformer networks (encoder and decoder) that follow an identical architecture. Each transformer network contains 6 layers, with a hidden dimension of 1536. Each layer is composed of a single-headed attention module with an attention head of size 64 followed by a 1-d convolutional block. Each convolutional block is a sequential operation of Conv1d, ReLU, Conv1d, Dropout and Layer Normalization. The kernel size for the convolution is 3 and dropout probability is 0.1.

The mel-spectrogram synthesizer also contains two submodules for predicting pitch and duration. The pitch and duration predictors take as input the output of the encoder network and predict a sequence of scalar values for duration or pitch (speaker normalized $F_0$ contour). Duration is used to regulate the length of the encoder output and pitch is embedded and concatenated with the encoder's output to be fed as input to the decoder. Both the pitch and duration predictor follow the same architecture — Each network contains two convolutional blocks. Each convolutional block is a serial composition of Conv1d, ReLU and layer normalization with a kernel size of 3 and hidden dimension of 256, followed by a linear layer that maps the hidden dimension to a scalar value for duration or pitch.

**HiFiGAN Vocoder:** The HiFi-GAN (Kong et al., 2020) vocoder used in this work consists of one generator and two discriminators: multi-scale and multi-period discriminators. In the generator network, consists of 4 upsampling blocks with an upsampling factor of 8, 8, 2, 2 with kernel sizes 16, 16, 4, 4 respectively. The model outputs audio at 22050Hz. The HiFiGAN vocoder is trained for 350 epochs on train-clean-360 subset of LibriTTS. Thereafter, the vocoder is additionally fine-tuned on synthetic mel-spectrograms, generated by the three mel-spectrogram synthesizers (Baseline-NoTransform, Baseline-Heuristic and SelfVC) for the same dataset for 5 epochs.

Table 4: Model size and wall clock inference time for a speech utterance of length 10 seconds using a batch size of 1 on CPU and NVIDIA RTX A6000 GPU.

| Model | # Parameters | Inference Time (Seconds) CPU | GPU |
|---|---|---|---|
| Speaker Encoder TiTaNet | 25 M | 0.13 | 0.05 |
| Conformer-SSL | 121 M | 0.44 | 0.10 |
| Mel-Spectrogram Synthesizer | 59 M | 0.15 | 0.01 |
| HiFiGAN Vocoder | 85 M | 2.1 | 0.08 |

### A.3 HEURISTIC TRANSFORMATION FUNCTIONS

For heuristic transformations, we follow the perturbation functions and hyperparameters proposed in (Choi et al., 2023). The three fundamental perturbation functions used are 1) Formant Shifting (fs) 2) Pitch Randomization (pr) and 3) Random Frequency Shaping (peq).

During training, the source utterance is perturbed by randomly choosing a transformation function $g_1$ or $g_2$ — Transformation function $g_1$ is a serial composition of *peq* and *fs*; And $g_2$ is a serial composition of *peq*, *pr* and *fs*.

For *pr*, pitch shift ratio is sampled uniformly from $U(1, 2)$ and pitch range ratio is sampled from $U(1, 1.5)$. Random frequency shaping (*peq*) is serial composition of low-shelfing, peaking and high-shelfing filters. Following NANSY, we use one low-shelving $H^{\text{LS}}$, one high-shelving $H^{\text{HS}}$, and eight peaking filters $H_1^{\text{Peak}}, \cdots, H_8^{\text{Peak}}$.

$$H^{\text{PEQ}}(z) = H^{\text{LS}}(z)H^{\text{HS}}(z)\prod_{i=1}^{8} H_i^{\text{Peak}}(z).$$

Each component is a second-order IIR filter parameterized by a cutoff/center frequency, quality factor, and gain parameter. The cutoff frequencies for $H^{\text{LS}}$ and $H^{\text{HS}}$ are set at $60Hz$ and $10kHz$, respectively. Center frequencies of $H_1^{\text{Peak}}, \cdots, H_8^{\text{Peak}}$ are uniformly spaced in between the shelving filters on a logarithmic scale. The quality factor of each component is randomly sampled as $Q = Q_{\min}(Q_{\max}/Q_{\min})^z$ where $Q_{\min} = 2$, $Q_{\max} = 5$, and $z \sim U(0, 1)$. The gain (in decibel) of each component is randomly sampled from $U(-12, 12)$.

We refer the readers to the link in the footnote (an unofficial open-source implementation of NANSY) for the precise implementation of transformation functions used in our work. [2]

### A.4 Voice Conversion on Seen Speakers and VCTK Datasets

We present results for additional experiments on speen speakers from train-clean-360 (using utterances from the hold out set) and unseen speakers from VCTK dataset in Table 5. We choose VCTK because it is an out-of-domain test set of unseen speakers for our models trained on LibriTTS. Similar to our primary experiments, we consider 20 source utterances, each from a different speaker and 10 target speakers resulting in 200 voice conversion trials. We compare against one prior work ACE-VC (Hussain et al., 2023), since ACE-VC is trained on the same dataset and VCTK dataset is not used during training. Other prior techniques considered in our main experiments conduct training on the VCTK dataset.

On the VCTK dataset, we find that SelfVC significantly outperforms the baselines and ACE-VC on the SV-EER metric. We also present the t-SNE plots for speaker embeddings of generated and real utterances in Figure 3. It can be observed that the embeddings of generated audio are closely clustered with the real embeddings of the target speaker for both seen and unseen speakers. We study the effect using different amounts of target speaker data when deriving speaker embedding for voice conversion in Figure 3. While the SV-EER improves as we incorporate more data from the target speaker, we observe marginal improvement beyond 16 seconds of target speaker data. In this graph, *seen speakers* refers to LibriTTS train-clean-360 and *unseen speakers* refers to VCTK.

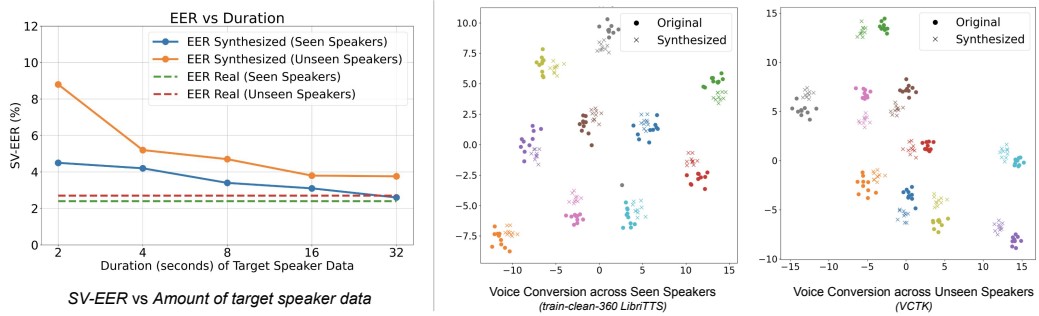

Figure 3: **Left**: SV-EER of voice converted speech generated by SelfVC using different amounts of target speaker data for estimating the speaker embedding. **Right:** t-SNE visualization of speaker embeddings of SelfVC synthesized and ground-truth audio for 10 target speakers. Each color represents a different speaker.

Table 5: Voice Conversion experiments on seen speakers (LibriTTS train-clean-360) and out-of-domain unseen speakers (VCTK). We compare against one prior work trained on the same dataset as ours.

| Technique | LibriTTS (train-clean-360) | | | VCTK | | |
| --- | --- | --- | --- | --- | --- | --- |
| | SV-EER ↓ | PER ↓ | CER ↓ | SV-EER ↓ | PER ↓ | CER ↓ |
| Real Data | 2.9% | 8.7% | 6.3% | 3.1% | 9.8% | 5.1% |
| ACE-VC Hussain et al. (2023) | 5.3% | 8.8% | 3.7% | 9.2% | 22.1% | 8.2% |
| Baseline–NoTransform | 19.1% | 5.5% | 2.6% | 25.2% | 7.6% | 3.8% |
| Baseline–Heuristic | 4.4% | 5.5% | 2.3% | 8.5% | 7.6% | 3.1% |
| SelfVC | **3.0**% | **5.4**% | **2.2**% | **4.3**% | 7.4% | **3.8**% |

---

[2] https://github.com/dhchoi99/NANSY/blob/master/datasets/functional.py

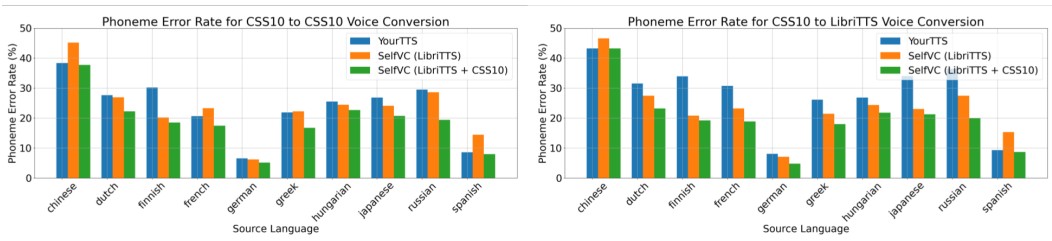

Figure 4: Phoneme Error Rate on Individual Languages of the CSS10 dataset for voice conversion experiments when the source utterance is from CSS10 and the target speaker is from another language in CSS10 or the LibriTTS test-clean dataset.

## A.5 MULTILINGUAL PHONEME ERROR RATE

In Figure 4 we present phoneme error rate on individual languages for *CSS10 to CSS10* and *CSS10 to LibriTTS* cross-lingual voice conversion experiments respectively. We also compare against the YourTTS model, which has the lowest average PER amongst the prior work considered in our work. As evident from the graphs, PER across all languages improve when SelfVC is fine-tuned on the LibriTTS train-clean-360 and CSS10 dataset (SelfVC (LibriTTS + CSS10) ). The fine-tuning is conducted for $10$ epochs with a fixed learning rate of $1e - 4$ on the combined LibriTTS and CSS10 dataset and takes around $5$ hours on a single NVIDIA RTX A6000 GPU. Certain languages such as Chinese, Russian and Japanese have higher PER across all methods. This is because of the large phonetic vocabulary of such languages which results in a higher PER from the wav2vec2 model even on real utterances (Xu et al., 2021).

## A.6 MOS AND SIM-MOS EVALUATION

**Naturalness MOS Evaluation:** We ask human listeners to rate the audio on a scale of $1$ to $5$ point naturalness scale with $1$ point increments. We present $200$ audio examples of each technique and each audio is independently rated by at least $4$ listeners. This results in a total of at least $800$ evaluations per technique. The template used for the Naturalness human study is shown in Figure 5. We report the MOS with $95\%$ confidence intervals in Table 2 of the paper.

**Speaker Similarity MOS (Sim-MOS):** For Sim-MOS evaluation, we ask human listeners to rate the speaker similarity of a given pair of utterances. For this evaluation, each synthetic utterance is paired with a real utterance of the target speaker. We create pairs for all of the $200$ synthesized utterances of each technique. Each pair is rated by at least $4$ independent listeners resulting in at least $800$ speaker similarity evaluations of each technique. We ask the listeners to judge only the voice/speaker of the utterances and ignore the accent, content, grammar and expressiveness of speech following past work (Jia et al., 2018; Casanova et al., 2022). The template used for this user study is shown in Figure 6. The Sim-MOS with $95\%$ confidence intervals in Table 2 of the paper. For reference, the reported Sim-MOS for same-speaker ground truth pairs is $4.36 \pm 0.08$ and different-speaker ground truth pairs is $1.77 \pm 0.10$.

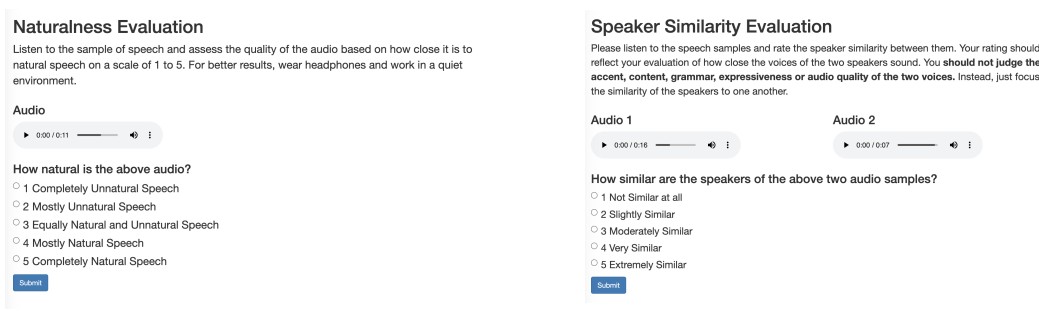

Figure 5: User Study template used for Naturalness MOS evaluation

Figure 6: User Study template used for Speaker Similarity MOS evaluation

