# OpenReview forum: "SelfVC: Voice Conversion With Iterative Refinement using Self Transformations"
_ICLR.cc/2024/Conference — Submitted to ICLR 2024_

### Official Review · Reviewer_PEBv · 2023-10-25

**Soundness:** 3 good
**Presentation:** 3 good
**Contribution:** 2 fair
**Rating:** 5
**Confidence:** 4

**Summary:**

The paper introduces SelfVC, a novel training strategy aimed at enhancing voice conversion models using self-synthesized examples. The proposed model integrates prosodic information from the audio signal for predictive training and uses a unique iterative training approach with self-synthesized examples for continuous model refinement. Compared to previous methods, SelfVC sets new SOTA in zero-shot voice conversion regarding naturalness, speaker similarity, and audio intelligibility.

**Strengths:**

1. The paper is well-composed, presenting its methodology with clarity.
2. The extensive experiments support the presented claims.
3. The demo provided by the author indicates the method's effectiveness.

**Weaknesses:**

Self-VC is similar to recent VC work (NANSY), except it uses pitch and duration predictors like ACE-VC. Also, as for the proposed training strategy (self transformations), random speaker embedding are commonly used for training a voice conversion model (e.g., https://arxiv.org/pdf/1806.02169.pdf, https://arxiv.org/pdf/2305.15816.pdf, https://arxiv.org/pdf/2305.07204.pdf,https://proceedings.neurips.cc/paper/2021/file/0266e33d3f546cb5436a10798e657d97-Paper.pdf). The fundamental idea seems the same. This point needs to be discussed more carefully.

**Questions:**

/

---

> ### Author Response · Authors · 2023-11-22
>
> Thank you for pointing out related work. We highlight the main differences with these works and provide clarification on the main contributions of our methodology below:
>
> Comparison with NANSY: NANSY does not offer duration and pitch control and only uses heuristic transformations during training. Our methodology not only improves speaker similarity over a model trained only with heuristic transformations, but also enables duration and pitch control by deriving duration information from SSL features in a text-free manner.
>
> Comparison with ACE-VC: While ACE-VC has duration and pitch predictors, ACE-VC relies on text transcripts to derive the duration targets for training the synthesizer model. In ACE-VC, SSL features corresponding to the same predicted character are grouped together. Requiring text transcripts is a severe limitation for scaling up the training data, including other languages or paralinguistic speech. In contrast, SelfVC derives the grouping is performed in a text-free manner using a cosine similarity based approach as described in Appendix A.1
>
> Our methodology enables the following:
>
> 1. Deriving content and speaker information with pitch and duration targets in a text-free manner. This allows training a controllable synthesis model that can adapt the prosody as per the target speaker embedding.
>
> 2. Iterative improvement of the voice conversion capabilities by challenging the model with self converted examples. Without requiring any adversarial loss or diffusion based model, our method proposes to derive the content representations from self-converted audios during training. This strategy iteratively improves the voice conversion capabilities as demonstrated by a significant improvement in speaker similarity metric across various voice conversion tasks and languages.
>
> 3. Our entire methodology is text-free and independent of phonetic posteriograms, hence making it simple and efficient to scale up the training data, including other languages.

---

### Official Review · Reviewer_YZVi · 2023-10-31

**Soundness:** 2 fair
**Presentation:** 3 good
**Contribution:** 2 fair
**Rating:** 5
**Confidence:** 4

**Summary:**

In this research, the authors introduce a new training strategy called SelfVC, aimed at enhancing voice conversion models by iteratively improving them using self-synthesized examples. While previous voice conversion efforts focused on separating speech attributes like speaker characteristics and linguistic content, this often results in the loss of finer details such as accents and emotions from the original audio. Instead of explicitly disentangling these attributes with loss terms, the authors propose a new approach that utilizes entangled speech representations derived from self-supervised learning and speaker verification models.

SelfVC framework comprises several key components. The authors introduce a training strategy that leverages self-synthesized examples to iteratively enhance the voice conversion model which is in contrast to NANCY or NANCY++. In this approach, the current state of the synthesis model is used to generate voice-converted versions of an utterance, which are subsequently used as inputs for the reconstruction task. This method ensures a continuous and purposeful refinement of the model.

The authors show that incorporating self-synthesized examples during training significantly improves the speaker similarity of the generated speech compared to a baseline voice conversion model trained solely on perturbed inputs. Since the framework does not rely on text, it can be applied to zero-shot voice conversion, voice conversion across different languages, and controllable speech synthesis with pitch and rhythm modifications.

The experiment section consist of matched and mismatched scenarion to give a better indication of the model's performance. In the matched setting, the authors evaluate the proposed model for speech reconstruction while in the mismatched scenario, speaker conversion is carried out.

**Strengths:**

The main strengths of the paper are:

The authors present a novel model focused on enhancing speaker conversion that operates independently of language and text. The core concept behind this model centers on iterative self-improvement. They address the challenges associated with the disentanglement of speaker and content attributes, which typically require auxiliary task training. Therefore, using self-improvement via iterative refinement provides a way to circumvent this disentanglement problem.

Their experiments conducted on the LibriTTS dataset reveal impressive reconstruction quality in both guided and predictive modes. Extending their investigations to CSS10 and VCTK datasets, the model demonstrates excellent performance in zero-shot scenarios and exhibits language-agnostic capabilities.

To evaluate their model, the authors employ a range of meaningful metrics, including CER, PER, SV-EER, and qualitative measures, providing comprehensive comparisons with several state-of-the-art models.

**Weaknesses:**

The main weakness of the work are as follows:

1. The proposed technique is relatively simple and falls short on the novelty axis. There are several works leveraging the idea of self-refinement which have been recently published such as:
(a) Self-Refine: Iterative Refinement with Self-Feedback - Madaan et. al.
(b) Meta Self-Refinement for Robust Learning with Weak Supervision - Zhu et. al.
(c) Safe Self-Refinement for Transformer-based Domain Adaptation - Sun et. al.

2. The authors mention that the proposed model is trained with fixed (pitch/formant) transformation for the first 100k steps and use self-refinement afterwards. A comparison of how the performance differs when the model is trained completely on deterministic transformation would strengthen the results.

**Questions:**

The authors should perhaps explain why in Table 1, 2 and 3, the PER is high on the real data but it goes down after processing through the voice conversion module. Is the overall pipeline achieving some sort of speech enhancement too?

---

> ### Author Response · Authors · 2023-11-22
>
> Thank you for the valuable feedback. We provide answers to the specific questions and clarifications to the comments below:
>
> Q. The authors should perhaps explain why in Table 1, 2 and 3, the PER is high on the real data but it goes down after processing through the voice conversion module. Is the overall pipeline achieving some sort of speech enhancement too?
>
> A. The PER for real data is the PER between the ground-truth transcription and the ASR transciption of real audio. The Real data PER is a reference to judge the accuracy of the ASR model. The PER of different speech synthesis techniques is between the ASR transcription of real audio and ASR transcription of generated audio. Therefore, the discrepancy between Real data PER and generated audio PER is expected.
>
> Comment: The authors mention that the proposed model is trained with fixed (pitch/formant) transformation for the first 100k steps and use self-refinement afterwards. A comparison of how the performance differs when the model is trained completely on deterministic transformation would strengthen the results.
>
> Clarification: We use 100k training iterations with heuristic transformations to train an initial voice conversion model that we can bootstrap for self-transformations. In our preliminary experiments, we found that using the initial heuristic transformations improve the convergence speed of training the model. We compare this model against a model trained solely on heuristic transformations and (Baseline–Heuristic) and provide the results in Table 2 and Table 3 of our paper.

---

### Official Review · Reviewer_vyxg · 2023-11-02

**Soundness:** 3 good
**Presentation:** 3 good
**Contribution:** 2 fair
**Rating:** 5
**Confidence:** 4

**Summary:**

* The paper focuses on zero-shot voice conversion and presents a framework for training a voice conversion model. The authors introduce a method that leverages entangled speech representations obtained from self-supervised learning (SSL) and speaker verification models. Additionally, they propose a novel training strategy that enhances the synthesis model for voice conversion through the use of self-synthesized examples.

**Strengths:**

* The paper combines strategies from voice conversion and singing voice conversion to improve zero-shot voice conversion. One noteworthy contribution is the introduction of a novel training strategy that utilizes self-synthesized examples for data augmentation and iterative improvement of the generation model. This represents a significant advancement from traditional approaches that heavily relied on heuristic transformations.

* The paper conducts comprehensive experiments. By extensively comparing the proposed framework with several baseline models, the authors effectively demonstrate its efficacy.

* Overall, the paper successfully presents a valuable contribution to the field of voice conversion and offers an approach for achieving zero-shot voice conversion.

**Weaknesses:**

* The main framework presented in this paper appears to draw from existing voice conversion and singing voice conversion techniques, such as utilizing SSL features, speaker embeddings, and incorporating prosody information like duration and pitch. These strategies resemble prior work ([1-3]) in the field. Additionally, the synthesizer's approach showcases similarities to methods used in speech synthesis, specifically resembling techniques employed in FastSpeech 2 [4]. To support these statements and provide a comprehensive overview of the related work in the field, it would be beneficial for the authors to provide appropriate citations and conduct comparative analyses.

* Moreover, the claim of the framework's efficiency in scaling to other languages through the introduction of SSL features is not unique to the authors' proposed model since similar approaches have been explored elsewhere.

* Additionally, it is worth noting that the concept of data augmentation using self-synthesized examples has been discussed in the literature, particularly in the context of speaker verification [5].

* Furthermore, the experimental results suggest that the inclusion of self-synthesized examples for data augmentation yields only a marginal improvement in the model's performance.

* Consequently, a deeper analysis or controlled study could be conducted to better isolate and understand the specific effects of the proposed training strategy from other contributing factors.

References:

[1] Jayashankar T, Wu J, Sari L, et al. Self-Supervised Representations for Singing Voice Conversion[C]//ICASSP 2023-2023 IEEE International Conference on Acoustics, Speech and Signal Processing (ICASSP). IEEE, 2023: 1-5.

[2] Hussain S, Neekhara P, Huang J, et al. ACE-VC: Adaptive and Controllable Voice Conversion Using Explicitly Disentangled Self-Supervised Speech Representations[C]//ICASSP 2023-2023 IEEE International Conference on Acoustics, Speech and Signal Processing (ICASSP). IEEE, 2023: 1-5.

[3] Maimon G, Adi Y. Speaking style conversion with discrete self-supervised units[C]//EMNLP 2023. EMNLP, 2023.

[4] Y. Ren, C. Hu, X. Tan, T. Qin, S. Zhao, Z. Zhao, and T.-Y. Liu, “Fastspeech 2: Fast and high-quality end-to-end text to speech,”[C]//International Conference on Learning Representations (ICLR), 2021.

[5] Cai D, Cai Z, Li M. Identifying Source Speakers for Voice Conversion Based Spoofing Attacks on Speaker Verification Systems[C]//ICASSP 2023-2023 IEEE International Conference on Acoustics, Speech and Signal Processing (ICASSP). IEEE, 2023: 1-5.

**Questions:**

* The authors should clarify what specific aspects of their framework, beyond the use of SSL features, contribute to this scalability to strengthen the claim of originality in this regard.

* Could the authors elucidate the unique characteristics or modifications they have made within their framework that differentiate it from other SSL-based voice conversion models?

* The introduction of the self-synthesized examples as a data augmentation method is intriguing. However, its efficacy and broader applicability remain questions. Can the authors provide experimental evidence or analysis showcasing the generalizability of this data augmentation technique across various methods?

---

> ### Author Response · Authors · 2023-11-22
>
> Thank you for your feeback and questions. We address the questions below:
>
> Q. The authors should clarify what specific aspects of their framework, beyond the use of SSL features, contribute to this scalability to strengthen the claim of originality in this regard.
>
> A. The main contribution of our paper is using self-synthesized training examples to improve voice conversion models by generating challenging training examples and iteratively improving the model. Besides, while prior work utilizes SSL features for voice conversion, they do not offer pitch and duration control over synthesized speech in a textless manner. Our approach allows duration and pitch control by deriving target duration and pitch in an unsupervised text-free manner during training and modeling the prediction in our synthesizer model. Finally, our approach does not require transcriptions, hence facilitating scaling up to larger datasets and more languages.
>
> Q. Could the authors elucidate the unique characteristics or modifications they have made within their framework that differentiate it from other SSL-based voice conversion models?
>
> A. We kindly point out that iterative model refinement using self-synthesized training examples (dubbed as Self-transformation in our paper) is a fundamentally different technique than self supervised learning. Our proposed technique for model refinement using self transformations is described in Section 3.3. The other main differentiator with past SSL based voice conversion work is text-free duration and pitch control as pointed above.
>
> Q. The introduction of the self-synthesized examples as a data augmentation method is intriguing. However, its efficacy and broader applicability remain questions. Can the authors provide experimental evidence or analysis showcasing the generalizability of this data augmentation technique across various methods?
> A. The concept of generating self-synthesized examples can be applied to analogous problems, where a controllable media synthesis model is trained with imperfectly disentangled representations. There have been prior efforts in using neural network generated examples as data-augmentation in the recognition domain [1][2]. In contrast, we propose a technique to use self-synthesized examples in iteratively improving a voice conversion model and demonstrate its efficacy in improving controllable synthesis models. We further demonstrate the efficacy of our proposed technique by achieving significant improvements across 10 different languages, as evidenced by the comprehensive evaluations provided in Table 3 and detailed further in Appendix A.5.
> We recommend future work in exploring the efficacy of this technique for tasks like image attribute manipulation and domain translation.
>
> [1] A simple framework for contrastive learning of visual representations, ICML, 2020
>
> [2] What makes for good views for contrastive learning?, NeurIPS, 2020

---

> > ### Comment · Reviewer_vyxg · 2023-11-23
> > **Thank authors for the responses**
> >
> > Thank the authors for the responses. I have also read the other reviewers' comments and authors' responses. I would like to raise my overall rating from 3 to 5.

---

### Meta-Review · Area_Chair_AB6p · 2023-12-04

**Metareview:**

The paper introduces SelfVC, a strategy for training voice conversion models using self-synthesized examples. This model incorporates prosodic information from the audio signal for predictive training and uses a unique iterative training method with self-synthesized examples for continuous refinement of the model. Despite achieving SOTA results in zero-shot voice conversion in terms of naturalness, speaker similarity, and audio intelligibility, the reviewers expressed concerns about the lack of sufficient novelty for acceptance.

**Justification For Why Not Higher Score:**

N/A

**Justification For Why Not Lower Score:**

N/A

---

### Decision · Program_Chairs · 2024-01-16

Reject